# DNA Damage Changes Distribution Pattern and Levels of HP1 Protein Isoforms in the Nucleolus and Increases Phosphorylation of HP1β-Ser88

**DOI:** 10.3390/cells8091097

**Published:** 2019-09-17

**Authors:** Soňa Legartová, Gabriela Lochmanová, Zbyněk Zdráhal, Stanislav Kozubek, Jiří Šponer, Miroslav Krepl, Pavlína Pokorná, Eva Bártová

**Affiliations:** 1Institute of Biophysics of the Czech Academy of Sciences, Královopolská 135, 612 65 Brno, Czech Republic; legartova@ibp.cz (S.L.); kozubek@ibp.cz (S.K.); sponer@ncbr.chemi.muni.cz (J.Š.); krepl@ibp.cz (M.K.); pokorna.pavlina@ibp.cz (P.P.); 2Central European Institute of Technology, Masaryk University, Kamenice 753/5, 625 00 Brno, Czech Republic; gabriela.lochmanova@ceitec.muni.cz (G.L.); zbynek.zdrahal@ceitec.muni.cz (Z.Z.)

**Keywords:** HP1, irradiation, nucleolus, phosphorylation, mass spectrometry, epigenetics, FLIM-FRET

## Abstract

The family of heterochromatin protein 1 (HP1) isoforms is essential for chromatin packaging, regulation of gene expression, and repair of damaged DNA. Here we document that γ-radiation reduced the number of HP1α-positive foci, but not HP1β and HP1γ foci, located in the vicinity of the fibrillarin-positive region of the nucleolus. The additional analysis confirmed that γ-radiation has the ability to significantly decrease the level of HP1α in rDNA promoter and rDNA encoding 28S rRNA. By mass spectrometry, we showed that treatment by γ-rays enhanced the HP1β serine 88 phosphorylation (S88ph), but other analyzed modifications of HP1β, including S161ph/Y163ph, S171ph, and S174ph, were not changed in cells exposed to γ-rays or treated by the HDAC inhibitor (HDACi). Interestingly, a combination of HDACi and γ-radiation increased the level of HP1α and HP1γ. The level of HP1β remained identical before and after the HDACi/γ-rays treatment, but HDACi strengthened HP1β interaction with the KRAB-associated protein 1 (KAP1) protein. Conversely, HP1γ did not interact with KAP1, although approximately 40% of HP1γ foci co-localized with accumulated KAP1. Especially HP1γ foci at the periphery of nucleoli were mostly absent of KAP1. Together, DNA damage changed the morphology, levels, and interaction properties of HP1 isoforms. Also, γ-irradiation-induced hyperphosphorylation of the HP1β protein; thus, HP1β-S88ph could be considered as an important marker of DNA damage.

## 1. Introduction

The chromatin of the eukaryotic cell nucleus is a dynamic and highly organized complex consisting of DNA and histones. In the interphase cell nucleus, chromatin appears as two primary and distinct isoforms referred to as euchromatin and heterochromatin. Generally, euchromatin is gene-rich, decondensed, replicates early in the S phase, and contains a histone post-translational modifications (PMTs), including histone acetylation, histone H3 lysine 4 di-/tri-methylation (H3K4me2/me3), or H3K36me2/me3, that are associated with active transcription. On the other hand, heterochromatin is gene-poor, highly condensed, and mostly remains transcriptionally silent throughout the cell cycle, replicates in the late S phase, which is the attribute of repetitive sequences. Constitutive heterochromatin preferentially consists of repetitive elements such as satellite DNA in centromeres or telomeric sequences [1,2,3]. In regions of constitutive heterochromatin, histones are mostly hypo-acetylated and, for example, hypermethylated on histone H3, lysine 9 (H3K9me2/me3). This type of histone post-translational modification represents a binding site for isoforms of heterochromatin protein 1 (HP1) [4,5,6,7].

Covalent modification of H3 at lysine 9 position is, in mammalian cells, catalyzed by a family of the SET-domain containing methyltransferases. The histone-lysine N-methyltransferase (SETDB1) and other enzymes, including SUV39H1 and SUV39H2 histone methyltransferases mediate both H3K9me2 and H3K9me3 [8,9]. The loss of the H3K9 methyltransferases (HMT) Suv39H1/H2 results in changes in HP1 localization [4]. Importantly, the two proteins, G9a (also known as EHMT2 or KMT1C) and GLP (G9a-like protein, also known as EHMT1 or KMT1D) catalyze H3K9me1 and H3K9me2 [10,11,12]. The histone H3K9me2/me3 modifications are well-known interacting partners of HP1, which can self-oligomerize. HP1 isoforms contribute to chromatin compaction [4,13,14]. Generally, the HP1 family of proteins plays an essential regulatory role in DNA-dependent processes, such as transcription, replication, and DNA repair. Moreover, especially HP1α protein is essential for the stabilization of heterochromatin.

The mammalian HP1 protein family contains three HP1 isoforms: HP1α, HP1β, and HP1γ that form homo- and heterodimers with each other [15]. All isoforms have specific domains that vary in their functions that were originally identified in *Drosophila melanogaster*, but these proteins are phylogenetically conserved and can be found in almost all eukaryotes, except for the budding yeast *Saccharomyces cerevisiae* [16]. The HP1 protein isoforms consist of two highly conserved domains. The one is an N-terminal chromodomain (CD), which specifically recognizes di- and trimethylated histones H3 (H3K9me2/me3) [4,14]. This interaction is essential for the recruitment of HP1 to heterochromatin [17,18]. On the other hand, the C-terminal chromo shadow domain (CSD) is responsible for dimerization of HP1 isoforms as well as for a wide variety of other protein-protein interactions [19]. For example, HP1 interacts with transcription regulators, chromatin modifiers, replication factors, cell cycle-related proteins, and components regulating nuclear architecture [20]. 

It is well-known that the CD and the CSD domains of HP1 are connected by a linker, respectively, the so-called hinge region (Hin). This region is of variable length and interacts with histones H1 or plays a role in the nonspecific binding of HP1 to DNA or RNAs [15,21,22]. The hinge region is highly opened for post-translational modifications, especially phosphorylation [23,24,25]. It has been shown that epigenetic modifications within this region affect localization, interactions, and function of HP1 isoforms [26]. In many cases, the function of HP1α, HP1β, and especially HP1γ do not entirely overlap, suggesting that these proteins can work independently [27,28,29]. 

It is well-known that HP1α, HP1β, and HP1γ are not only localized to different regions of the cell nucleus [30], but these proteins are also characterized by different dynamics in differentiated cells and during cell cycle progression [27,31,32]. However, the significance of HP1 specific localization is not well understood. Generally, it is thought that HP1 isoforms independently regulate multiple functions in the genome [33]. From the view of localized kinetics, Cheutin [34] showed that HP1 recovery kinetics after photobleaching correlate with the degree of chromatin condensation. For example, the recovery kinetics after photobleaching was slower for HP1α accumulated in heterochromatin when compared with HP1α mobility in euchromatic regions [34]. Legartová [35] additionally documented that in apoptotic cells, mobility of HP1α is maintained in later stages of this process which was different, for instance, from GFP-tagged Jmjd2b histone demethylase that was immobile in the apoptotic cell nucleus.

Later on, Yearim [36] showed that HP1 also regulates alternative splicing of pre-mRNA in DNA methylation-dependent manner. In this epigenetic regulatory pathway, HP1 mediates a direct effect of DNA methylation on splicing, which is a novel and very rare observation of how epigenetic events take part in splicing machinery [36,37]. Above mentioned data showed how HP1 isoforms are multifunctional due to the fact that these proteins can regulate not only processes of heterochromatinization and gene silencing, but also splicing or apoptosis. 

Recently, it was observed that HP1β and HP1γ play an essential role in the regulation of ribosomal gene transcription [38,39,40]. Yuan [39] showed that HP1γ binds to a complex consisting of Cockayne syndrome group B (CSB) protein, RNA polymerase I (RNA Pol I), and H3K9me2. Importantly, in the absence of CSB, the RNA Pol I is not able to associate with ribosomal DNA (rDNA). Moreover, the CSB deficiency also reduces the interaction between RNA Pol I and G9a methyltransferase. Thus, the synthesis of the pre-rRNA in the nucleolus is inhibited [39]. In the compartment of nucleoli of mouse embryonic fibroblasts, Horakova [40] additionally observed the co-localization between HP1β and RNA Pol I-positive regions. This co-localization property presupposes that also HP1β is essential for the transcription of ribosomal genes. Moreover, Horakova et al. [40] showed that this process was independent on the function of SUV39h histone-methyltransferases (Suv39h1 and Suv39h2 in the mouse genome); thus, there is no direct link to the function of H3K9me3.

Also, it is well-known that the HP1 isoforms bind not only to the methylated H3K9 but also to several non-histone proteins [26]. For example, all three isoforms of mammalian HP1 protein can create a complex with the KRAB-associated protein 1 (KAP1, also known as TRIM28 or TIF1β). This protein was identified as a universal transcriptional co-repressor because it interacts with a Krüppel-associated box zinc finger protein (KRAB-ZFP) [41]. The interaction between KAP1 and HP1 is mediated via well-defined PxVxL peptide motif, and the mutagenesis of this motif abolishes the ability of KAP1 to silence transcription [42]. Cammas [43] showed that in the undifferentiated cells, KAP1 has a diffuse pattern, but in the differentiated cells, the KAP1 protein relocalized to foci of pericentromeric heterochromatin where KAP1 colocalized with HP1. However, the cells with mutation in HP1-binding domain of KAP1 fail to colocalize with the HP1 protein during differentiation. This observation supports the idea that KAP1 interaction with HP1 is required for its integration into the higher-order heterochromatin [43,44]. In Bártová [45] we additionally showed that nuclear arrangement of HP1 isoforms and TIF1β (syn. KAP1, TRIM28) is distinct in various differentiation pathways, induced in embryonal carcinoma cells. However, the levels of HP1 isoforms and TIF1β were relatively stable in non-differentiated cells when compared with their differentiated counterpart. Interestingly, during the cell cycle, in late G1, and early S-phase of the cell cycle, KAP1 is phosphorylated at serine 473, and there is the highest level of phosphorylation of KAP1-S473 [46]. Chang [47] hypothesized that the presence of this phosphorylation near the HP1-binding domain on KAP1 might affect its association with HP1 isoforms. Hiragami-Hamada [48] showed that N-terminal phosphorylation of HP1α as essential for its binding to chromatin. Their biochemical analyses revealed that HP1α was multiply phosphorylated at N-terminal serine residues (S11–14) in human and mouse cells and that this phosphorylation enhanced HP1α’s affinity for H3K9 methylation. 

To contribute to this knowledge, here, we address a question of how phosphorylation of HP1β can be changed by γ-irradiation and how radiation affects dimerization between HP1 isoforms. From this view, it is well-known that HP1 isoforms play a role in DNA damage response, as it was published by other authors [49,50]. Moreover, we recently showed that hyperacetylation, induced by inhibitors of histone deacetylases, abrogates recruitment of HP1β to microirradiated chromatin [51], and HP1β participates in Nucleotide Excision Repair (NER) recognizing UVA-induced cyclobutane pyrimidine dimers (CPDs) [52]. Here, we also address a question of how γ-irradiation can change the distribution pattern and levels of HP1 isoforms localized as foci in vicinity of nucleoli. Our observation could contribute to the knowledge of how HP1 isoforms regulate DNA damage response and how irradiation by γ-rays changes HP1 properties in the nucleolus. 

## 2. Materials and Methods

### 2.1. Cell Cultivation and Treatment

The human cervix adenocarcinoma (HeLa) cell line (ATCC^®^ CCL-2™) were cultivated in Eagle’s Minimum Essential Medium (EMEM) supplemented with 10% of fetal bovine serum (FBS) and appropriate antibiotics at 37 °C in a humidified atmosphere containing 5% CO_2_. 24 h after seeding the HeLa cells were irradiated with either 2 Gy or 5 Gy of γ-rays with cobalt-60 (Chisostat, Chirana, Czech Republic) as the irradiation source. Cells were also treated with HDAC inhibitor (HDACi) Suberoylanilide Hydroxamic Acid (SAHA, #149647-78-9, Cayman Pharma, Czech Republic) at the final concentration 15 μM. The cells were harvested 10 min and 30 min after irradiation with 2 Gy, two hours after irradiation with 5Gy or two hours after treatment with HDACi and fixed for further analysis by western blots, immunofluorescence, immunoprecipitation (IP), and confocal microscopy. Also, we study an effect of 2 Gy of γ-ray and its combination with SAHA treatment. In this case, we analyzed two intervals (2 h after SAHA treatment in combination with 2 Gy of γ-irradiation; cells were harvested 10 min or 30 min after irradiation).

For transient transfection, HeLa cells were cultivated on uncoated 50-mm glass-bottomed dishes, used for inverted microscopy (#P50G-1.5-30-F, No. 1.5; MatTek Corporation, USA), to 70% confluence, and transfected with 2 μg or 5 μg plasmid DNA encoding GFP-HP1α (#17652 Addgene, USA; [34]), GFP-HP1β (#17651 Addgene, USA; [34]) and GFP-HP1γ (#17650 Addgene, USA; [34]). Dr. Oskar Laur (Emory University School of Medicine, Atlanta, USA) performed re-cloning and inserted the CBX1 gene sequence into the mCherry-pBABE puro vector (#1764 Addgene, USA; [53]) Dr. Martijn S. Luijsterburg (Department of Cell and Molecular Biology, Karolinska Institute, Stockholm, Sweden) generously provided plasmids encoding GFP-HP1β-ΔCSD, GFP-HP1β-ΔCD, and GFP-HP1β-ΔHinge [50]. Transfections were performed using METAFECTENE™ PRO reagent (#T040-2.0, Biontex Laboratories GmbH, Germany), and concentrations were optimized for individual plasmids. For plasmid amplification, we used chemically competent *E. coli* DH5α bacteria, and plasmid DNA was isolated with the QIAGEN Plasmid Maxi Kit (#121693, Qiagen, Bio-Consult, Czech Republic).

### 2.2. Immunofluorescence

Cells were fixed in 4% formaldehyde for 10 min at room temperature. Permeabilization of the cell nuclei was performed by incubation with 0.2% Triton X-100 (#194854, MP Biomedicals, USA) for 10 min and 0.1% saponin (#S7900, Sigma Aldrich, Czech Republic) for 12 min. Next, the slides were washed twice in phosphate-buffered saline for 15 min. The 1% bovine serum albumin (BSA; #A2153-506, Sigma Aldrich, Czech Republic) dissolved in PBS was used as a blocking solution. The samples were incubated in a blocking buffer for one hour and then washed for 15 min in PBS. For immunofluorescence analysis, we used the following antibodies: HP1α (#05-689, Merck, Czech Republic), HP1β (#MAB3448, Merck, Czech Republic), HP1γ (#05-690, Merck, Czech Republic), KAP1 (#ab10483, Abcam, UK), anti-histone H2AX phosphorylated at the S139 position (γH2AX; #ab2893, Abcam, UK), 53BP1 (#MAB3802, Merck, Czech Republic), and fibrillarin (#ab4566, Abcam, UK). The primary antibodies were diluted 1:100 in PBS containing 1% BSA. As secondary antibodies, we used Alexa Fluor^®^594-conjugated donkey anti-rabbit IgG (H + L) (#A21207, Thermo Fisher Scientific Inc., USA), Alexa Fluor^®^488-conjugated goat Anti-Rabbit IgG (H&L) (#ab150077, Abcam, UK), Alexa Fluor^®^594-conjugated goat anti-mouse IgG (H + L) (#A11032, Thermo Fisher Scientific Inc., USA), for STED microscopy we used secondary antibodies Abberior STAR 580 (#2-0012-005-8, Abberior GmbH, Germany) and Abberior STAR 635P (#2-0002-007-5, Abberior GmbH, Germany). The secondary antibodies were diluted 1:200 in PBS containing 1% BSA. 

The DNA content was visualized using 4′,6-diamidino-2-phenylindole (DAPI; #D9542, Sigma-Aldrich, Czech Republic), and Vectashield (#H-1000, Vector Laboratories, USA) was used as the mounting medium. 

Additionally, we analyzed the area and number of γH2AX foci in non-irradiated cells, cells irradiated by 5 Gy of γ-rays (cells were harvested 2 h after irradiation), 2 Gy of γ-rays (cells were harvested 30 min after irradiation), or 2 Gy of γ-rays (cells were harvested 30 min after irradiation) combined with SAHA treatment. We additionally studied the volume and the number of γH2AX- and 53BP1-positive DNA repair in above-mentioned samples. The analysis was performed by the use of ImageJ software (NIH freeware) which enables to automatically find the individual DNA damage related foci and their number, and volume.

### 2.3. Confocal Microscopy, Image Analysis and STED Microscopy

We acquired images with Leica TCS SP8 X SMD confocal microscope (Leica Microsystem, Germany). Image acquisition was performed using a white light laser (WLL) with the following parameters: 1024 × 1024-pixel resolution, 400Hz, bidirectional mode, and zoom 8–12. For 3D projections, we obtained 30–40 optical sections with the axial step of 0.3 μm. Reconstruction of 3D projection was conducted using the Leica Application Suite (LAS) software.

Stimulated emission depletion (STED) microscopy was performed on the TCS SP8 STED 3X (Leica Microsystems) equipped with 660 nm and 775 nm STED lasers, STED White Objective CS 100 ×/1.40 OIL. Image acquisition was performed using depletion laser of 775 nm and 3X–3D STED, gating 0.3–6 ns for both Abberior STAR 580 and Abberior STAR 635P detection. Images were acquired using LAS X software and deconvolved by Huygens Professional software.

### 2.4. Immunoprecipitation

To investigate protein–protein interactions, cells were grown to 70% confluence and then 24 h after seeding cells were washed in cold PBS buffer and incubated in Pierce^TM^ IP Lysis Buffer (#87788, Thermo Fisher Scientific Inc., USA), supplemented with protease and phosphatase inhibitor cocktail (#78440, Thermo Fisher Scientific Inc., USA) for 5 min on ice. The total protein concentration was determined by the DC protein assay kit (#5000111, Bio-Rad, Bio-Consult, Czech Republic) and by ELISA Reader μQuant (BioTek, USA). IP was performed according to the manufacturer’s protocol (Catch and Release^®^v2.0 Reversible Immunoprecipitation System, #17-500, Merck, Czech Republic). This kit contains a proprietary resin in a microfuge-compatible Spin Column secured by a screw cap top and a breakaway closure on the bottom. The binding between the resin and antigen: antibody complex is mediated via Antibody Capture Affinity Ligand. As the first step in the protocol the Spin Columns with resin were washed twice with 1× Wash Buffer, after that the reagents were added at Spin Columns in the following order: 1× Wash Buffer, Cell lysate, Specific primary antibody or negative control antibody (IgG whole molecule, #A4914, Sigma-Aldrich, Czech Republic) and Antibody Capture Affinity Ligand. IP reactions were performed overnight at 4 °C. The next day Spin Columns were washed three times with 1× Wash Buffer and proteins were eluted from Columns by 1× Denaturing Elution Buffer containing β-Mercaptoethanol (to a final concentration of 5%). Precipitates were fractionated by sodium dodecyl sulfate-polyacrylamide gel electrophoresis (SDS-PAGE) of western blot. For the IP experiments, we used the following antibodies: HP1α (#2616S, Cell Signaling Technology, The Netherlands), HP1β (#39979, Active Motif, Belgium), HP1γ (#05-690, Merck, Czech Republic) and KAP1 (#ab10483, Abcam, UK). 

### 2.5. Western Blotting

Western blot analysis was performed following [35]. Briefly, protein loading was calibrated to achieve identical concentrations of total protein. Protein concentration was measured with a µQuant spectrophotometer (BioTek, USA). Then the proteins were separated by SDS polyacrylamide gel electrophoresis (SDS-PAGE) and transferred to polyvinylidene difluoride (PVDF) membranes. The membranes were blocked with 2% nonfat milk or 2% gelatin for 1 h and then immunoblotted with various antibodies overnight at 4 °C. We used the following primary antibodies: HP1α (#2616S, Cell Signaling Technology, The Netherlands), HP1β (#39979, Active Motif, La Hulpe, Belgium), HP1γ (#05-690, Merck, Czech Republic), KAP1 (#ab10483, Abcam, UK), anti-H3K9 acetylation (#06-942, Merck, Prague, Czech Republic), anti-histone H3 phosphorylated at the S10 antibody (H3S10ph, #ab5176, Abcam, Cambridge, UK) and anti-histone H2AX phosphorylated at the S139 position (γH2AX, #ab2893, Abcam, Cambridge, UK). The western blot data were normalized to the amount of total protein, total histone H3 (#ab1791, Abcam, Cambridge, UK) and α-tubulin (#LF-PA0146, Thermo Fisher Scientific Inc., Waltham, MA, USA). Primary antibodies were diluted 1:1000–1:3000; as secondary antibodies, we used goat anti-rabbit IgG (#AP307P, Merck, Prague, Czech Republic; 1:2000), rabbit anti-mouse IgG (#A9044, Sigma-Aldrich, Czech Republic; 1:2000), and goat anti-mouse IgG1 (#sc-2060, Santa Cruz Biotechnology, Dallas, TX, USA; 1:2000). The western blot data (density of western blot bands) were quantified by ImageJ software (NIH freeware), and Student’s *t*-test was used for statistical analysis (Sigma Plot software v14.0, Jandel Scientific, San Jose, CA, USA). Statistical significance at *p* ≤ 0.05 is shown by (*).

### 2.6. Fluorescence Lifetime Image–Förster Resonance Energy Transfer (FLIM-FRET) Technique

Fluorescence Lifetime Image (FLIM) Microscopy combined with Förster Resonance Energy Transfer (FRET) was performed following [54]. By using FRET-FLIM technique, we studied the interaction between GFP-HP1α and mCherry-HP1β, GFP-HP1β and mCherry-HP1β, and finally GFP-HP1γ and mCherry-HP1β. According to Reference [55], we selected as the well-known interacting partner with isoforms of protein HP1, the KAP1 protein. First, we measured a decrease of fluorescence lifetime of the donor (τ_D_) in fixed cells after immunostaining and then a decrease of fluorescence lifetime of the donor in the presence of acceptor (τ_DA_). In our experiments, we used transiently transfected HeLa cells with GFP-HP1α, GFP-HP1β, and GFP-HP1γ (donor-only). For donor-acceptor pairs we either co-transfected into the cells GFP-HP1α, β or γ with mCherry-HP1β or performed visualization of KAP1 by immunostaining (Alexa 594-visualized KAP1). For FLIM-FRET measurements, we used Leica TCS SP8 X SMD confocal microscope (Leica Microsystems GmbH, Germany), supplemented by PicoHarp 300 module (PicoQuant GmbH, Germany) and HyD SMD detectors. A 63× oil immersion objective of 1.4 numerical aperture was used for photon acquisition. As the excitation source, we used the pulsed white-light laser (WLL, 470–670 nm in 1-nm increments) with a repetition rate of 20 MHz. We acquired at least 500 photons/ pixel at resolution 512 × 512 pixels. We analyzed results by SymPhoTime 64 software (PicoQuant GmbH, Germany), FRET efficiency was calculated as mean fluorescence lifetimes weighted by amplitudes according to the formula [56,57]:$$FRET\;Efficiency=\frac{\left(1/r\right)6\;}{\left(1/r\right)6+\;k_{other}}\;=\;\frac{R_{0}^{6}}{R_{0}^{6}+r^{6}}\;=1-\;\frac{\tau_{DA}}{\tau_{D}}$$
Ten cell nuclei were studied by FRET-FLIM technique for each experimental event.

### 2.7. Immunoaffinity Enrichment of HP1β for mass Spectrometric Analysis

Transiently transfected HeLa cells with plasmid DNA encoding GFP-tagged HP1β (#17651 Addgene, USA; [34]) were lysed with RIPA buffer supplemented with Phosphatase Inhibitor Cocktail 2 (#P5726, Sigma-Aldrich, Prague, Czech Republic), 1mM PMSF, and 45mM sodium butyrate. GFP-HP1β fusion protein was immunoprecipitated using ChromoTek GFP-Trap^®^ (#gtma-20, ChromoTek, Planegg-Martinsried, Germany) according to manufacturer instructions. The immunocomplex was washed four times, with 500 µL of 50 mM ammonium bicarbonate buffer (ABC). On-bead digestion was performed using 20 ng·µL^−1^ Glu-C endoproteinase in 50 µL of 50 mM ABC. The procedure was performed overnight in thermomixer at 37 °C and 1100 rpm. The beads were magnetically separated, followed by an additional 5 h incubation at 37 °C and 750 rpm. The samples were acidified, and the sample volume was reduced in a Savant SPD121P concentrator to 10 µL (Thermo Fisher Scientific Inc., Waltham, MA, USA).

### 2.8. Mass Spectrometric Analysis

Glu-C digests of three batches represented by independent biological replicates were measured using liquid chromatography tandem mass spectrometry (LC-MS/MS). The samples were spiked with the iRT-C18 reference peptides (iRT-Kit, #Ki-3002-1, Biognosys AG, Schlieren, Switzerland). The LC-MS/MS equipment consisted of an RSLCnano system, equipped with an X-Bridge BEH 130 C18 trap column (3.5 µm particles, 100 µm × 30 mm; Waters, Milford, MA, USA), and an Acclaim PepMap100 C18 analytical column (3 µm particles, 75 µm × 500 mm; Thermo Fisher Scientific Inc., Waltham, MA, USA), coupled to an Orbitrap Lumos Tribrid spectrometer (Thermo Fisher Scientific Inc., Waltham, MA, USA) equipped with a Digital PicoView 550 ion source (Scientific Instrument Services, Ringoes, NJ, USA), and Active Background Ion Reduction Device. Prior to LC separation, Glu-C digests were online concentrated on trap column. The mobile phase consisted of 0.1% formic acid in water (A) and 0.1% formic acid in 80% acetonitrile (B), with the following proportions of B: 1% for 3 min at 500 nL/min, then with a switch to 300 nL/min for next 2 min, increased linearly from 1 to 30% over 70 min, 30–56% over 35 min, 56–80% over 5 min and followed by isocratic washing at 80% B for 10 min. Equilibration with 99:1 (mobile phase A:B; flow rate 500 nL/min) of the trapping column and the column was done prior to sample injection to sample loop. The analytical column outlet was directly connected to the ion source of the MS. MS data were acquired using a data-dependent strategy with 3 s cycle time based on precursor abundance in a survey scan (350–2000 *m/z*). The resolution of the survey scan was 60,000 (400 *m/z*) with automatic gain control (AGC) target value of 4 × 10^5^, one microscan and maximum injection time of 50 ms. The selection of precursors for MS/MS mode occurred in the quadrupole. HCD MS/MS spectra were acquired with an AGC target value of 5 × 10^4^ and resolution of 30,000. The maximum injection time for MS/MS was 500 ms. Dynamic exclusion was enabled for 60 s after one MS/MS spectrum acquisition. The isolation window for MS/MS fragmentation was set to 1.6 *m/z*. 

### 2.9. Database Searches and Quantification of HP1β Peptide Forms

The RAW mass spectrometric data files were analyzed using Proteome Discoverer software (Thermo Fisher Scientific, USA; version 2.2) with in-house Mascot search engine (Matrixscience, UK; version 2.6.2) to compare acquired spectra with entries in the UniProtKB human protein database (version 2018_09; 21053 protein sequences), cRAP contaminant database and in-house HP1 database (version 2019_05; 6 protein sequences). Mass tolerances for peptides and MS/MS fragments were 10 ppm and 0.02 Da, respectively. For the HP1 database searches, Glu-C enzyme specificity with up to six missed enzyme cleavages and the following modifications were set: acetylation (K, protein N-term), methylation (K, R), dimethylation (K), trimethylation (K), and phosphorylation (S, T, Y) as variable modifications. For searches against the cRAP and UniProtKB human databases, Glu-C enzyme specificity with up to two missed enzyme cleavages, and oxidation (M), and deamidation (N, Q) as variable modifications were set. Rank 1 peptides with Mascot expectation value < 0.01 and Ion Score ≥ 30 were considered. The abundance of peptides was quantified automatically using Proteome Discoverer 2.2 software. The peak intensity corresponding to each precursor ion was calculated from the extracted ion chromatograms (XICs) using the Precursor Ions Quantifier node. Selected HP1β peptide identifications were manually verified and quantified from the peak areas derived from the XICs using Skyline 4.2 software, including identification alignment across the raw files based on retention time and *m/z*. The mass spectrometry proteomics data, shown here, have been deposited to the ProteomeXchange Consortium via the PRIDE partner [58] repository with the dataset identifier PXD014802.

### 2.10. ChIP-Polymerase Chain Reaction (ChIP-PCR) Analysis of HP1 Isoforms in Ribosomal Genes

ChIP-polymerase chain reaction (PCR) was performed following the protocol published by Strašák et al. [59]. Briefly, the DNA with the associated proteins was cross-linked for 15 min at 37 °C by adding formaldehyde (#252549, Sigma-Aldrich, Prague, Czech Republic), to a final concentration of 1%, directly to the cell culture medium. The medium was removed, and the cells were washed twice in ice-cold PBS containing protease inhibitor cocktail (#87786, Thermo Fisher Scientific Inc., USA). The following steps were performed according to the manufacturer’s protocol ChIP assay Kit (#17-295, Merck, Prague, Czech Republic). The HeLa cells (1–2 × 10^6^ per sample) were lysed in 1% SDS lysis buffer containing protease inhibitor cocktail, followed by incubation on ice for 10 min. Lysates were sonicated twice, 5–6 times for 10 s. To verify sonication efficiency, the samples were subjected to agarose gel electrophoresis. Optimal fragment sizes ranged from 200 to 1000 bp. The samples used for further analyses were centrifuged at 20,000 rcf for 10 min at 4 °C. In the next step, 900 μL of ChIP dilution buffer was added to 100 μl of sonicated lysate. We used following primary antibodies: HP1α (#05-689, Merck, Czech Republic), HP1β (#MAB3448, Merck, Czech Republic), HP1γ (#05-690, Merck, Czech Republic), KAP1 (#ab10483, Abcam, UK) and negative control (IgG whole molecule, #A4914, Sigma-Aldrich, Czech Republic). The next day, samples were incubated with 12 μg of Protein A agarose/salmon sperm DNA (50% Slurry) for two hours at 4 °C and centrifuged at 200 rcf for 2 min. The agarose beads with antigen: antibody complexes were washed using ChIP assay Kit buffers. Each wash was performed for 5 min with rotation and was followed by centrifugation at 200 rcf at 4 °C, for 2 min. The immune complexes were eluted from the agarose beads by incubating samples twice with 250 μL elution buffer (1% SDS and 0.1M NaHCO_3_) for 15 min at RT with rotation, and then supplemented with 20 μL of 5 M NaCl. To reverse cross-links, samples were incubated overnight at 65 °C. The next day, the DNA was isolated using the QIAamp DNA Mini Kit (#51304, Qiagen, Bio-Consult, Prague, Czech Republic). All analyses included input DNA samples, which originated from a portion (2%) of lysates obtained before IP. The primers for rDNA promoter and rDNA encoding 28S rRNA were adapted following [40,60]. PCR reactions were performed in a Peltier Thermal Cycler (Bio-Rad, Hercules, USA). PCR products were separated by electrophoresis on 2% agarose gels and visualized using GelRED (#41003; Biotium, Fremont, CA, USA). The density of ChIP-PCR fragments was measured by ImageQuant LAS 3000 software (GE Healthcare, Pittsburgh, PA, USA) using the approach described by Jugova et al. [61] or by Morgenstern and Greenland [62].

## 3. Results

### 3.1. Distribution Pattern of HP1 Isoforms in the Compartment of the Nucleolus

By the use of confocal microscopy, we studied nuclear distribution pattern of HP1 isoforms, including HP1α, HP1β, and HP1γ. We addressed a question if irradiation by γ-rays changes the morphology of HP1-positive foci and their localization close to nucleoli (Figure 1a–d). Except for HP1α, we observed non-significant changes in the distribution of HP1 isoforms in HeLa cells when exposed to 5 Gy of γ-rays. However, in the case of HP1α, γ-radiation reduced the number of HP1α-positive foci in the vicinity of nucleoli (Figure 1a,d). These results we also confirmed quantitatively by ChIP-PCR analysis showing a reduced level of HP1α in both rDNA encoding 28S rRNA and rDNA promoter region (Figure 2a,b). ChIP-PCR additionally showed, in comparison to HP1α and HP1β, the highest level of HP1γ in ribosomal genes, and especially in rDNA promoter regions. Importantly, γ-irradiation did not affect the level of HP1γ in ribosomal genes (Figure 2a,b).

### 3.2. Mass Spectrometry Showed an Increased Level of HP1β Ser88 Phosphorylation Induced by γ-Irradiation

GFP-tagged HP1β protein was immunoprecipitated from the non-irradiated control samples, γ-irradiated (5 Gy), and SAHA-treated HeLa cells. Cell originated from three independent cultivations, and for IP, we used ChromoTek GFP-Trap^®^ kit. Among more than 800 identified proteins, GFP-tagged HP1β was found to be the most abundant protein with magnitude higher level than other proteins, and the ~75% sequence coverage. In HP1β sequence, following post-translational modifications (acetylation and phosphorylation) were identified with high confidence: K83ac, K85ac, S88ph/S90ph, S161ph/Y163ph, S171ph, and S174ph. The most PTMs were detected in the peptide sequences corresponding to the hinge region and CSD domain while no PTM was detected within the CD domain of the HP1β protein (Figure 3).

To examine the influence of γ-rays and SAHA treatment on HP1β post-translational modifications, the abundance of following phosphorylated peptides was quantified (Figure 4a–e): G81GKRKADS88phDSEDKGEE96, A151NVKCPQVVISFY163phEE165, R166LTWHS171phYPSE DDDKKDDKN184, R166LTWHSYPS174phEDDDKKDDKN184, R166LTWHS171phYPS174phE DDDKKDDKN184. Compared to the non-irradiated control samples, significantly increased level of G81–E96 peptide carrying phosphorylation at serine 88 (HP1β-S88ph; *m/z* 596.5863+++) was observed in γ-irradiated samples (Figure 4a and Figure 5a,b). Although more than a 1.5-fold increase in the level of this peptide was also found in the cells treated by clinically used HDAC inhibitor SAHA, the data did not fulfill the limit of p-value to be considered as significant (at *p* < 0.05; Figure 4a and Figure 5a). Except for G81GKRKADS88phDSEDKGEE9 peptide, the abundance of other above-mentioned phosphopeptides was not influenced by either γ-rays or SAHA treatment (Figure 4b–e). Due to the results showing an increased HP1β-S88ph in γ-irradiated HeLa cells, and these cells treated by SAHA (Figure 4a), we addressed a question if HDAC inhibitor, SAHA, induces DNA damage. In the cells treated by HDACi, we did not observe a multiplication of 53BP1-positive DNA repair foci, as it is caused by γ-rays (Figure 6a–e). However, the morphology and volume of 53BP1-foci in SAHA-treated cells (1–2 foci per cell nucleus) was distinct from spontaneously occurring DNA repair foci (Figure 6a–e). We can summarize that HDAC inhibition, due to hyperacetylation effect, causes decondensation of 53BP1-foci and remarkable γH2AX (phosphorylation of histone H2AX) positivity in the nucleoplasm (Figure 6a,e). In comparison with ionizing-irradiation induced foci (IRIF), caused by γ-rays, the damaging effect of SAHA on the genome was insignificant. From the view of HP1β-S88ph, we can summarize that this is the post-translation modification of HP1β that seems to be sensitive to both irradiation and hyperacetylation, a process with a potential to change chromatin structure.

Although searching for HP1β-binding partners by mass spectrometric analysis was not a primary goal of our study, we noticed that several known interaction partners or potential candidates were isolated together with the bait protein. In particular, endogenous HP1α and HP1γ isoforms were identified by LC-MS/MS. However, no PTMs were detected within their sequences. Further, KAP1, a protein previously described to be essential for physiological functions of HP1β, appeared within the top 30 identified proteins (see Appendix A). Based on this observation, we addressed a question if nuclear distribution pattern and a degree of interaction are identical for HP1α-KAP1, HP1β-KAP1, and HP1γ-KAP1. 

### 3.3. Protein–protein Interactions: H1β Forms Homodimers and This Interaction is Mediated via CSD Domain. Interaction Between HP1β and KAP1 is Strengthened by HDAC Inhibitor

Here, as a reference interaction partners for FLIM-FRET analysis, we used HP1 and KAP1 proteins (Figure 7(a)a). By FLIM-FRET we observed a pronounced interaction between HP1α-KAP1 and weaker interaction between HP1β and KAP1 (Figure 7(a)b). We also confirmed these results by IPs (Figure 8a,b). By the use of both methods, we revealed no interaction between HP1γ and KAP1 (Figure 7(a)c, Figure 8c). An interaction between HP1 isoforms and KAP1 was not affected by γ-irradiation (Figure 8a–c). In the cell nuclei, analyzed by FLIM-FRET, we observed a co-localization between HP1α-KAP1, HP1β-KAP1, and HP1γ-KAP1, but compartment of nucleoli was abundant in HP1β-, and HP1γ-positive foci, not co-localizing with KAP1 (Figure 7a,b). STED analysis showed a relatively high level of the KAP1 protein inside nucleoli decorated by HP1β and HP1γ (Figure 7c). However, ChIP-PCR analysis showed the absence of KAP1 in rDNA encoding 28S rRNA a relatively low level of the KAP1 protein we only found in rDNA promoter regions (Figure 2a,b). 

According to the FLIM-FRET analysis, we revealed additional information: in HeLa cells, the interaction between HP1β and KAP1 was surprisingly mediated via the Hinge region of HP1β, as shown in Figure 7(d)a–c. For explanation, in the case of GFP-tagged HP1β ΔCD/Alexa 594-KAP1 and GFP-tagged HP1β ΔCSD/Alexa 594-KAP1 we observed FLIM-FRET efficiency approximately (11–14)%, however, for GFP-tagged HP1β ΔHinge/Alexa 594-KAP1, it was (6.3 ± 1.1)%, which implies an importance of the hinge region of HP1β in HP1β-KAP1 interaction (Figure 7(d)a–c). 

Here, we also studied the effect of SAHA treatment and γ-irradiation, as well as the effect of their combination (Figure 8d,e). Interestingly, SAHA treatment did not influence levels of HP1α, HP1β, HP1γ, KAP1, and γH2AX, but only H3S10ph was reduced 2 h after HDAC inhibition (see the inserted panel in frame shown in Figure 8d). Similarly, γ-irradiation induced H3S10 dephosphorylation, in parallel with an increase in γH2AX (Figure 8d). Importantly, SAHA treatment, combined with γ-irradiation, increased the levels of HP1α and HP1γ, which was accompanied by a pronounced H3S10 phosphorylation (Figure 8d,e). Both γ-irradiation alone or combination of 2 Gy with SAHA treatment increased the number of γH2AX-positive foci when compared with non-treated cells; an average area of these DNA repair foci was significantly enlarged when HeLa cells were irradiated by 5 Gy of γ-rays (Figure 9). 

By FLIM-FRET analysis, we also studied the degree of interaction between HP1 isoforms. We observed the following interacting partners: heterodimer HPα-HP1β, homodimers HP1β-HP1β, and week bindings we found for HP1β-HP1γ (Figure 10(a)a–c). According to FLIM-FRET results, HP1β-HP1β dimerization is mediated via chromoshadow domain (CSD) of HP1β, as we showed by the use of deletion mutants (GFP-tagged HP1β ΔCSD, GFP-tagged HP1β ΔCD, GFP-tagged HP1β ΔHinge). Shortly, a relatively high FLIM-FRET efficiency we observed when chromodomain (CD) of HP1β and the hinge region of HP1β were deleted, which support our conclusion that it is CSD of HP1β that is essential for HP1β dimerization (Figure 10(b)a–c). 

FLIM-FRET data we compared with IP results. By IPs, we observed the following interactions: HP1α-HP1α, but not HP1α-HP1β or HP1α-HP1γ. Inhibition of histone deacetylases by SAHA strengthened HP1α homodimerization and interaction between HP1β and KAP1 (Figure 8a,b asterisks). According to IPs, HP1β interacts with HP1γ, and HP1γ form dimers (Figure 8c). Interestingly, γ-irradiation did not affect mentioned protein–protein interactions. Similarly, the global levels of HP1 isoforms were not significantly changed by both doses of γ-rays (2 Gy and 5 Gy) (Figure 8d,e). 

## 4. Discussion

By mass spectrometry, we observed 8 sites in the HP1β protein carrying post-translational modifications (GKKQNKKKVE EVLEEEEEEY VVEKVLDRRV VKGKVEYLLK WKGFSDEDNT WEPEENLDCP DLIAEFLQSQ KTAHETDKSE GGK*RK*ADS#DS# EDKGEESKPK KKKEESEKPR GFARGLEPER IIGATDSSGE LMFLMKWKNS DEADLVPAKE ANVKCPQVII S#FY#EERLTWH S#YPS#EDDDKK DDKN), including #phosphorylation and *acetylation. While the presence of S88ph, S90ph, and S174ph was previously detected in the phosphoproteomic studies dealing with different cell types (i.e., HeLa S3 cells, hESC lines, liver cells) [56,57,58], phosphorylation at sites S161 / Y163 have not been described as yet according to our knowledge. Importantly, S171ph was found in breast cancer and HeLa cell lines (https://www.phosphosite.org/siteAction.action?id=6208903). Modified sites observed, here, in HeLa cells, were different from those observed by LeRoy et al. [55] providing proteomic analysis of HP1α, HP1β, and HP1γ in HEK cells. In our study, phosphorylation was found as prevailing PTM of the HP1β protein, isolated from HeLa cells, whereas no phosphosite within HP1β sequence was identified in LeRoy’s study. Post-translational modification status of HP1 isoforms was also analyzed by Lomberk et al. [63]. These authors observed that the phosphorylation of HP1γ at serine 83 is essential for HP1γ binding to euchromatin, in which HP1γ interacts with DNA repair factors [63]. Recently, HP1β phosphorylation at Thr-51 has been shown to be important during the initiation step of the DNA damage response. The more functional explanation is following: during DNA damage HP1β was phosphorylated at threonine-51 (HP1β-T51ph) which was mediated by casein kinase 2 (CK2) and this process influenced H2AX phosphorylation that is considered as a significant epigenetic marker of damaged chromatin [49]. From the view of DNA repair, here, we add new information about DNA damage-related functions of HP1β. We showed that HP1β-S88ph is a striking marker that is significantly changed in the γ-irradiated genome of HeLa cells (Figure 4a). In these cells, we did not observe HP1β-T51ph (Figure 3), as it was detected by Ayoub et al. [49] in U2OS cells. Importantly, among five phosphorylated sites observed in our samples, HP1β-S88ph was the only one PTM of the HP1β protein that was significantly increased in HeLa cells exposed to 5 Gy of γ-rays (Figure 4a and Figure 5a,b).

Also, because we noticed a high abundance of KAP1 protein in HP1β fraction, and as it was published by Lechner et al. [64], we analyzed which domains of HP1 isoforms are essential for HP1 interaction with the KAP1 protein and HP1 dimerization (Figure 7a–d and Figure 10(a)a–(b)c). Using GFP-tagged HP1β ΔCD/Alexa 594-KAP1 and GFP-tagged HP1β ΔCSD/Alexa 594-KAP1 we revealed FLIM-FRET efficiency (11–14)%, however, for GFP-tagged HP1β ΔHinge/Alexa 594-KAP1, FRET efficiency was (6.3 ± 1.1)%, which implies that not only CSD, as published by Lechner et al. [64], but also a part of the hinge region could mediate HP1β interaction with KAP1 (Figure 7d). Importantly, Lechner et al. [64] showed that CSD (15-amino-acid segment) of HP1α (data were not provided for HP1β) is a specific repression domain interacting directly with KAP1. Interestingly, here, we observed a new phenomenon that hyperacetylation induced by SAHA strengthened the interaction between HP1β and KAP1, and also dimerization of HP1α as it was shown by IPs (Figure 8a,b, asterisks).

As the next step, we compared our results from IPs with FLIM-FRET data, and we can only state that results are not identical, especially in the case of HP1 dimerization. Putting together data from IPs and FLIM-FRET, we summarize that in HeLa cells, there is an existence of following protein-protein interactions: HP1α-HP1α, HP1α-KAP1, HP1β-HP1β, HP1β-KAP1, HP1α-HP1β, HP1β-HP1γ, HP1γ-HP1γ. However, by any method, we did not observe an interaction between HP1α-HP1γ, and interaction between HP1γ-KAP1. This data confirmed a distinct role of HP1γ in chromatin remodeling when compared to HP1α or HP1β and their function in heterochromatinization and gene expression, as it was shown by many authors [4,65,66,67]. We additionally found a high density of HP1γ in ribosomal genes, and by this result, we confirmed data published by [39] showing that HP1γ, in parallel with G9a-mediated H3K9me2, regulates transcription of ribosomal genes [39]. 

As mentioned above, isoforms of the HP1 protein has distinct functions, although they are characterized by relatively identical nuclear distribution pattern (Figure 1a–c, [4,32,40]). Here, we show that γ-irradiation has the ability to reduce the number of HP1α-positive foci in the vicinity to nucleoli, and the level of HP1α in ribosomal genes was also diminished in the cell exposed to γ-rays. Interestingly, nucleoli are decorated by all HP1 isoforms, but both HP1β and HP1γ foci on the nucleolar periphery did not co-localize with the KAP1 protein. These results confirm our claim mentioned above that distinct nuclear distribution patterns or co-localization profiles of various proteins can indicate an existence of their distinct functions. For example, we recently showed that HP1α and HP1β do not co-localize with 53BP1-positive DNA repair foci appearing in the vicinity to the nucleoli [68]. This observation indirectly implied that studied HP1 isoforms do not participate in 53BP1-mediated NHEJ repair mechanism, as it was shown later on by Zarebski et al. [69] or Stixova et al. [52]. 

## 5. Conclusions

Together, in HeLa cells, we showed that γ-irradiation reduced the level of HP1α in close proximity to nucleoli, and HP1β-S88 is prone to phosphorylation that is increased in the genome exposed to γ-rays. Thus, in tumor cells like HeLa, HP1β-S88ph could be considered as a marker of DNA damage. Importantly, DNA damage does not affect dimerization of HP1 isoforms and HP1-KAP1 interactions, but surprisingly, HDACi strengthened interaction between HP1β and KAP1 proteins. Importantly, the combination of both HDACi treatment and irradiation by 2 Gy of γ-rays increased the global level of HP1α and HP1γ; thus, it likely affects the function of HP1 isoforms in DNA damage response. 

## Figures and Tables

**Figure 1 cells-08-01097-f001:**
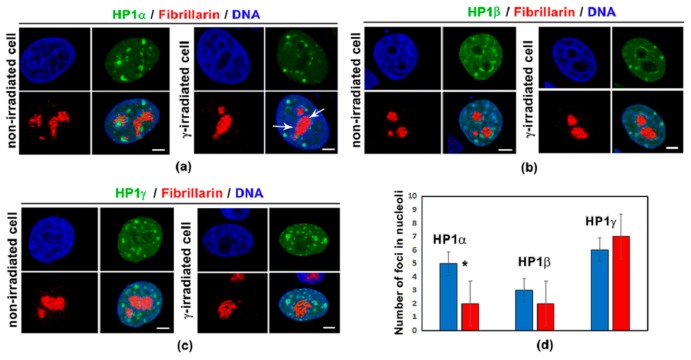
Nuclear distribution pattern of heterochromatin protein 1 (HP1) isoforms in non-irradiated and γ-irradiated human cervix adenocarcinoma (HeLa) cells. Morphology of (**a**) HP1α (see GFP-tagged HP1α, green), (**b**) HP1β (see GFP-tagged HP1β, green), and (**c**) HP1γ (see GFP-tagged HP1γ, green) was studied by laser scanning confocal microscopy. The HP1-positive foci (green) were analyzed in the vicinity of fibrillarin-positive regions of nucleoli (red fluorescence signals). DAPI staining (blue) was used for visualization of nuclei. Scale bars show 5 µm. (**d**) Analysis of the number of HP1α-, HP1β-, and HP1γ-positive foci in non-irradiated and γ-irradiated (5 Gy) cells. Cells were fixed for analysis 2 h after γ-irradiation. Fifty cell nuclei per sample were studied. Asterisk in panel (d) shows significantly different result.

**Figure 2 cells-08-01097-f002:**
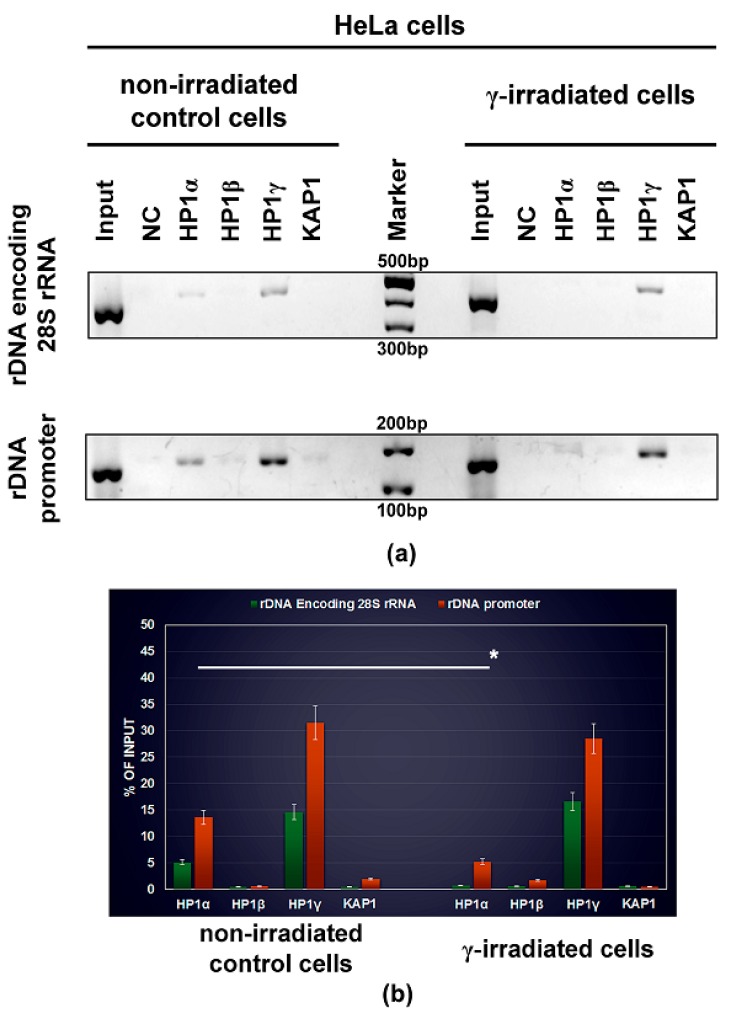
ChIP-polymerase chain reaction (ChIP-PCR) analysis of HP1α, HP1β, HP1γ, KAP1 abundance in ribosomal genes. (**a**) The analysis was performed in non-treated human cervix adenocarcinoma (HeLa) cells, and cells irradiated by 5 Gy of γ-rays. Cells were harvested 2 h after γ-irradiation. The highest density in rDNA encoding 28S rRNA and rDNA promoter region was observed for HP1γ. (**b**) Quantification of fragment densities, shown in panel (**a**), was done by ImageJ software. White asterisk shows statistically significant difference, shown by Student’s t-test at *p* ≤ 0.05.

**Figure 3 cells-08-01097-f003:**
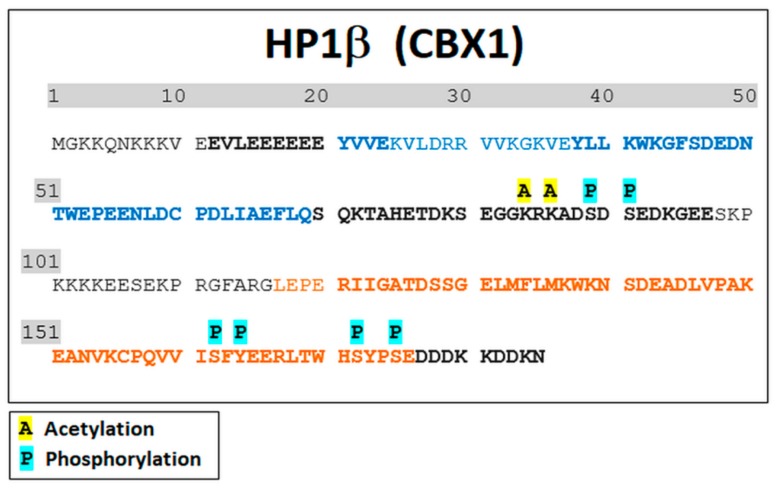
Characterization of HP1β protein in human cervix adenocarcinoma (HeLa) cells using MS. Glu-C endoproteinase was used for protein digestion. After filtering the LC-MS/MS data in Proteome Discoverer 2.2 (see Methods for details), 75% sequence coverage was obtained (marked in bold type). Identified PTMs, including phosphorylation (P) and acetylation (A) are indicated. The amino acid sequence corresponding to CD and CSD domain is shown in blue or orange, respectively, with hinge region in between.

**Figure 4 cells-08-01097-f004:**
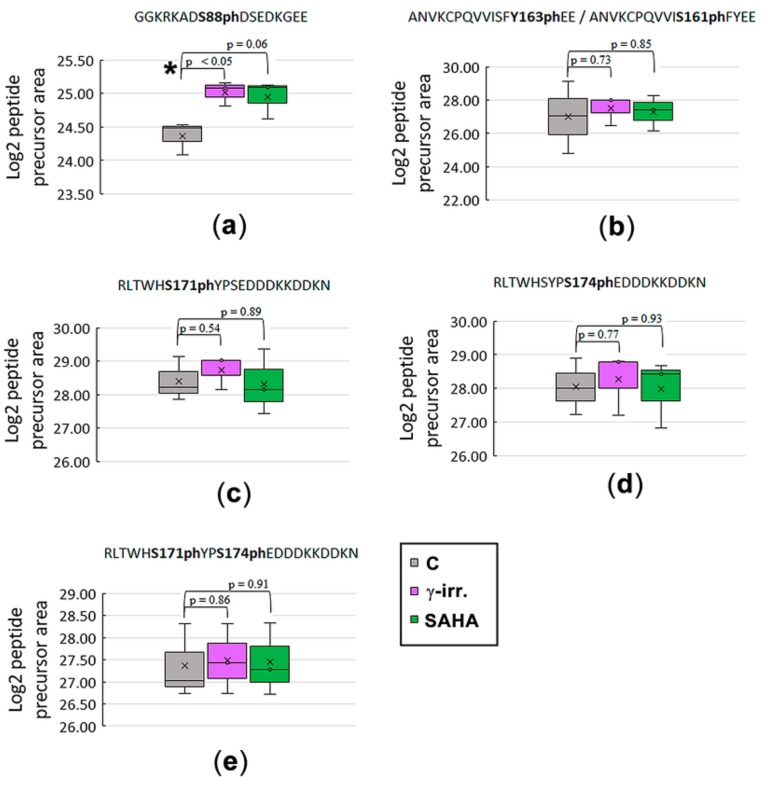
The proportion of signature marks on HP1β protein in non-treated, γ-irradiated, and SAHA treated human cervix adenocarcinoma (HeLa) cells. Box-plots of the relative abundance of particular phosphopeptides showing extremes, interquartile ranges, means and medians (*N* = 3). Precursor peak of phosphopeptides was quantified in Skyline SW. Data were normalized to the sum of selected non-modified peptides. Following post-translational modifications were analyzed: (**a**) HP1β-S88ph; (**b**) HP1β-Y163/S161ph; (**c**) HP1β-S171ph; (**d**) HP1β-S174ph; (**e**) HP1β-S171ph-S174ph. Differences between samples in normalized peptide abundances ≥1.5-fold were considered as significant, and the significance of differences was assessed using Student’s t-tests, setting the significance threshold at *P* < 0.05 (shown by the asterisk). Cells were harvested 2 h after the treatment; 5Gy of γ-rays or 15 µM SAHA.

**Figure 5 cells-08-01097-f005:**
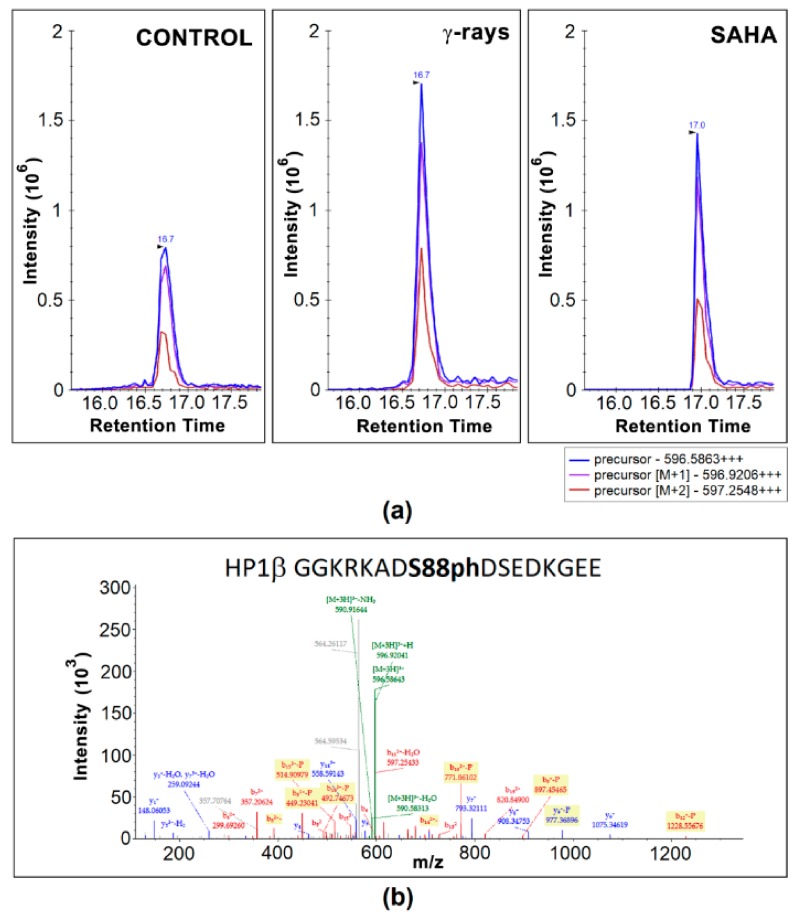
Representative MS and MS/MS spectra of HP1 G81−E96 peptide carrying phosphorylation at S88 (*m/z* 596.586). (**a**) MS extracted ion chromatograms of the 3+ charged HP1β-S88ph peptide precursor showing changes in its abundance after γ-irradiation (5 Gy) and SAHA (15 μM) treatment. The three colored lines correspond to the ion chromatogram of the monoisotopic peptide mass and the first two isotopes. (**b**) MS/MS spectrum produced from the precursor ion of *m/z* 596.586. The deposition of the mass spectrometry proteomics data see at the ProteomeXchange Consortium (for detailed information, see Methodology section).

**Figure 6 cells-08-01097-f006:**
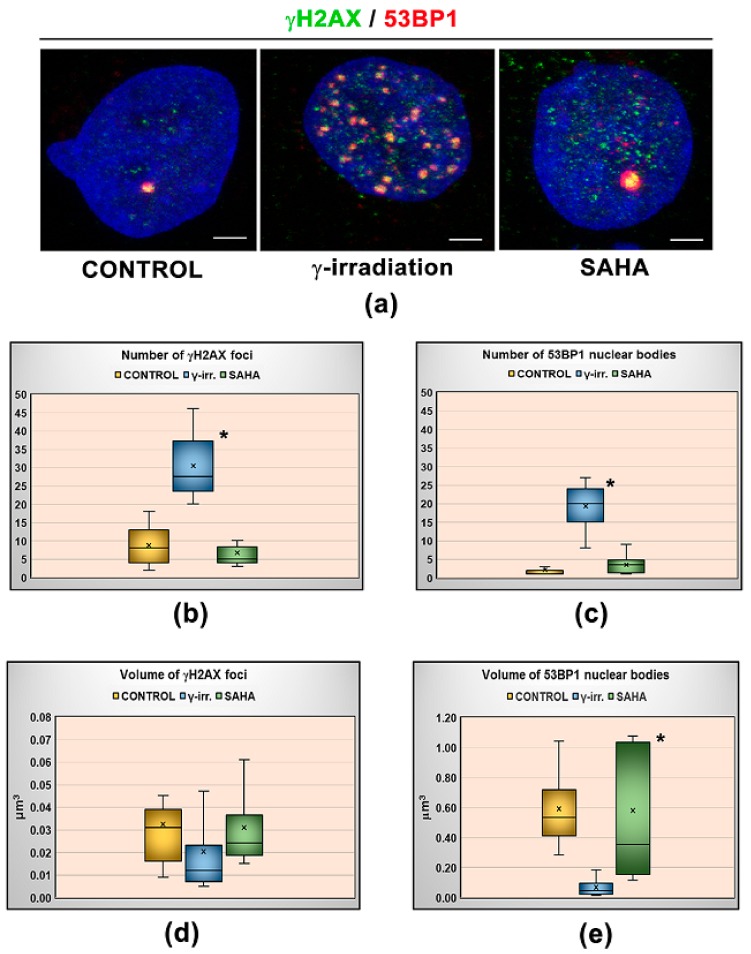
Nuclear distribution pattern of 53BP1 foci and phosphorylated histone H2AX. (**a**) Close proximity or co-localization between 53BP1 foci (red) and phosphorylated histone H2AX (γH2AX, green) in control, γ-irradiated (5 Gy) and SAHA (15 μM) treated human cervix adenocarcinoma (HeLa) cells. Cells were harvested 2 h after the treatment. Scale bars show 5 µm. Irradiation by γ-rays increased the number of (**b**) γH2AX and (**c**) 53BP1-positive foci. (**d**) The volume of γH2AX foci in control, γ-irradiated (5 Gy) and SAHA (15 μM) treated cells remain unchanged. (**e**) SAHA treatment enlarged the volume of 53BP1-positive foci, as indicated by asterisk showing a large scale of 53BP1-foci volume. Irradiation by γ-rays reduced the number of 53BP1-positive foci (blue box). Immunofluorescent data were quantified using the ImageJ software. Fifty cell nuclei per sample were studied. The box plots are displaying the distribution of data based on the summary of five numbers: minimum, first quartile, median, third quartile, and maximum. Asterisks show significantly different results from control values, at *p* ≤ 0.05. Student’s t-test was used for data analysis.

**Figure 7 cells-08-01097-f007:**
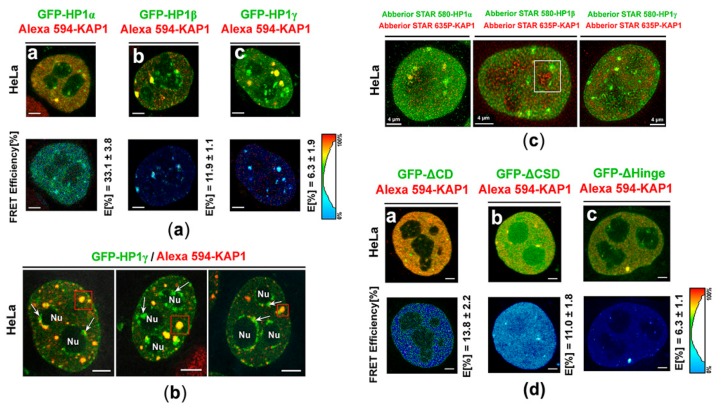
Fluorescence Lifetime Imaging-Forster Resonance Energy Transfer (FLIM-FRET) analysis of potential interaction between heterochromatin protein 1 (HP1) protein isoforms and the KAP1 protein. (**a**) Analysis of a GFP-tagged HP1α-KAP1, b] GFP-tagged HP1β-KAP1, and c] GFP-tagged HP1γ-KAP1. (**b**) An example of a nuclear distribution of HP1γ and the KAP1 protein. HP1γ-positive regions of nucleoli (green) were absent of KAP1. Abbreviation Nu means nucleoli; arrows show HP1γ foci absent of KAP1 and red frames show HP1γ foci colocalizing with KAP1. (**c**) Example of STED analysis showing the location of KAP1 inside the cell nucleoli decorated by the HP1 isoforms (see HP1β in white frame). (**d**) FLIM-FRET analysis of a] GFP-tagged HP1β ΔCD/Alexa 594-KAP1, b] GFP-tagged HP1β ΔCSD/ Alexa 594-KAP1, and c] GFP-tagged HP1β ΔHinge/Alexa 594-KAP1. E [%] means FLIM-FRET efficiency. Scale bars represent 4 µm.

**Figure 8 cells-08-01097-f008:**
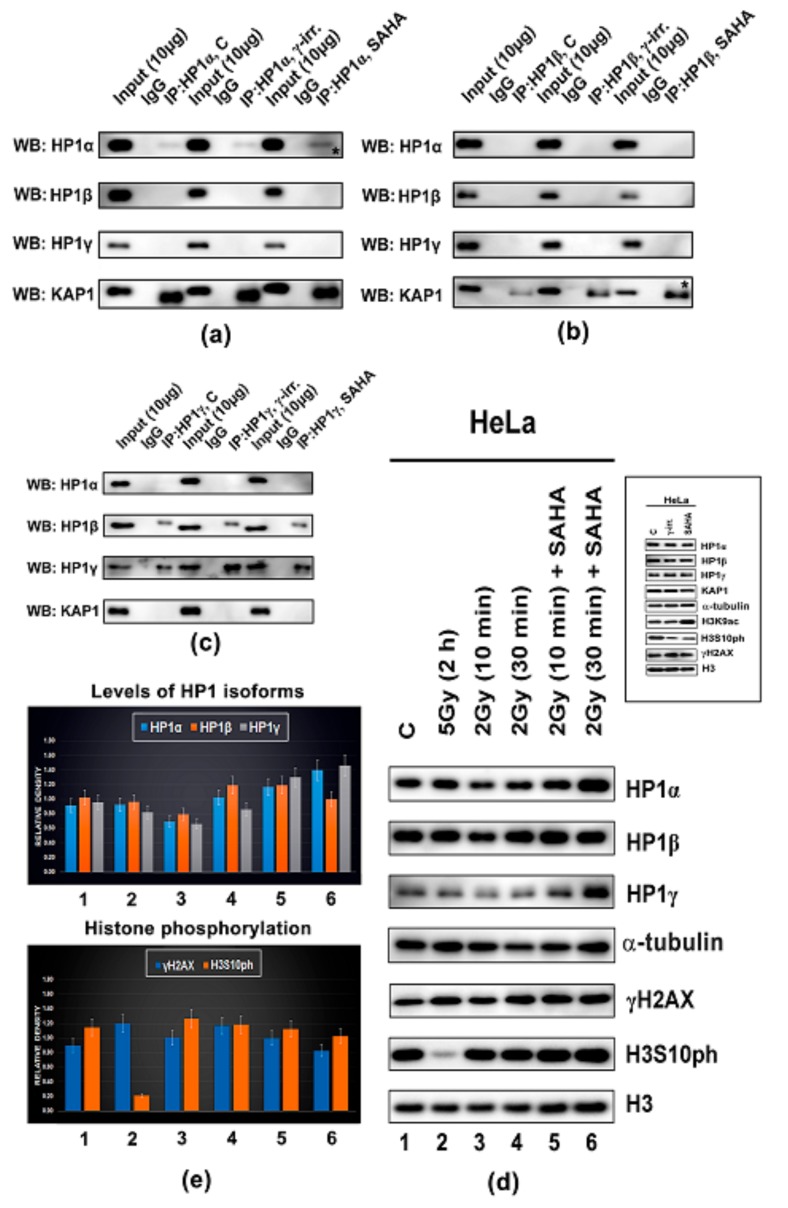
Immunoprecipitation analysis of potential interaction between HP1α, HP1β, and HP1γ. Studies of following possible interactions: (**a**) HP1α-HP1α, HP1α-HP1β, HP1α-HP1γ; HP1α-KAP1; (**b**) HP1β-HP1α, HP1β-HP1β, HP1β-HP1γ; HP1β-KAP1; (**c**) HP1γ-HP1α, HP1γ-HP1β, HP1γ-HP1γ, HP1γ-KAP1. Protein-protein interaction was studied in non-treated HeLa cells, and cells treated with SAHA (15 µM) and irradiated by γ-rays (5 Gy). Cells were harvested for analysis 2 h the treatment. (**d**) Western blot analysis of following proteins: HP1α, HP1β, HP1γ, KAP1, normalized to the total protein levels and α-tubulin, and H3K9ac, H3S10ph, γH2AX, normalized to the level of total histone H3. (**e**) Data from large panel (**d**) were quantified by ImageJ software. Levels of HP1 isoforms were normalized to the level of α-tubulin, and γH2AX or H3S10ph levels were normalized to the level of total histone H3. Profiles of HP1α, HP1β, HP1γ, KAP1, H3S10ph, and γH2AX were studied in (no. 1) non-irradiated HeLa cells, γ-irradiated Hela cells by (no. 2) 5 Gy of γ-rays/harvested after 2h; (no. 3) 2 Gy/harvested after 10 min; (no. 4) 2 Gy/harvested after 30 min; (no. 5) combination of SAHA treatment with 2 Gy/harvested after 10 min; and (no. 6) combination of SAHA treatment with 2 Gy/harvested after 30 min. A small panel in (**d**) is showing western blot data for non-treated HeLa cells, and cells irradiated by 5 Gy of γ-rays or SAHA (15 µM) treated HeLa cells.

**Figure 9 cells-08-01097-f009:**
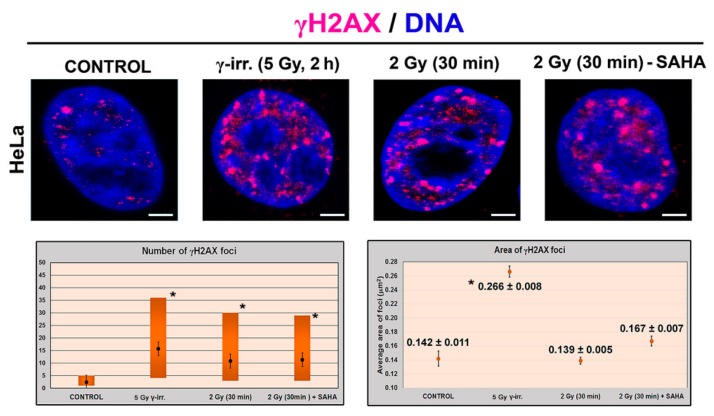
Effects of γ-irradiation by 5 Gy of γ-rays, SAHA treatment, and a combination of 2 Gy with SAHA treatment. The number and area of γH2AX-positive foci were measured. Data were compared with non-treated cells. Immunofluorescent data were quantified using the ImageJ software. Fluorescence intensity (FI) was analyzed, and foci with FI ≥ 100 were used for analysis. Number of repair foci are shown as a minimal and maximal value, and area of γ-positive foci is shown as mean ± standard errors (S.E.). Asterisks show statistically significant results at *p* ≤ 0.05. Student’s t-test was used for statistical analysis. Scale bars show 5 µm.

**Figure 10 cells-08-01097-f010:**
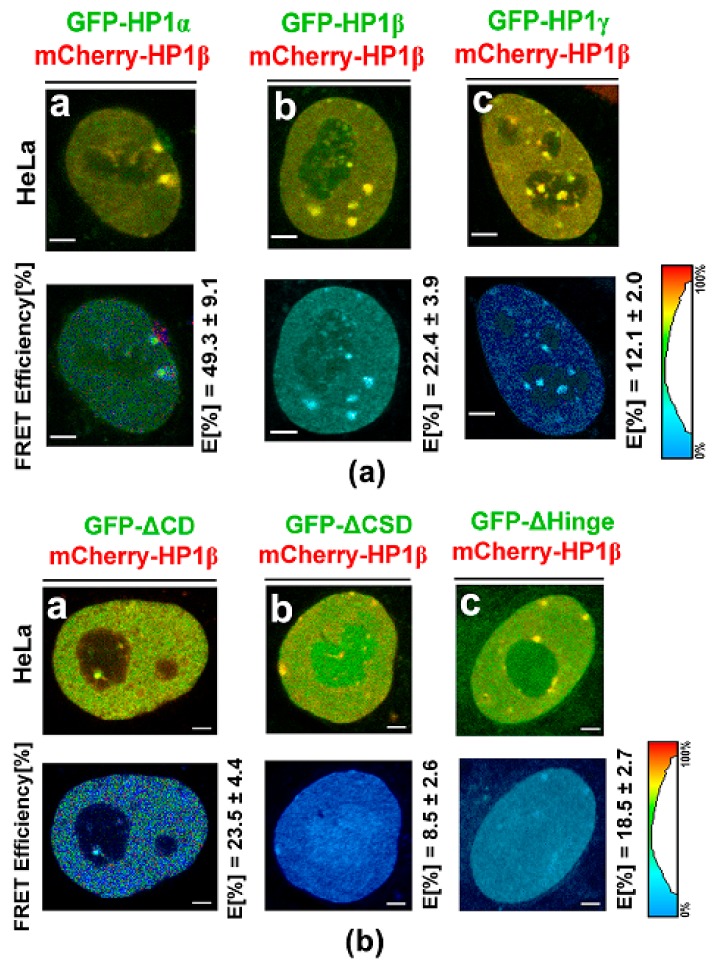
Fluorescence lifetime imaging-Forster resonance energy transfer (FLIM-FRET) analysis of potential interaction between HP1α, HP1β, and HP1γ. Following interactions were studied: (**a**) a) GFP-tagged HP1α/mCherry-HP1β, b) GFP-tagged HP1β/ mCherry-HP1β, **c]** mCherry-HP1β/GFP-tagged HP1γ. E (%) means FLIM-FRET efficiency. FLIM-FRET analysis of (**b**) a) GFP-tagged HP1β ΔCD/mCherry-HP1β, b) GFP-tagged HP1β ΔCSD/mCherry-HP1β, and **c**) GFP-tagged HP1β ΔHinge/mCherry-HP1β. E (%) means FLIM-FRET efficiency. Scale bars represent 2 µm.

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
