# Peer review of "DNA Damage Changes Distribution Pattern and Levels of HP1 Protein Isoforms in the Nucleolus and Increases Phosphorylation of HP1β-Ser88"

_cells, 2019, doi:10.3390/cells8091097_

Round 1
Reviewer 1 Report
The manuscript by Legartova S et al. explores the relationship between sensing/accumulation of DNA Damage and the regulation of chromatin environment, focussing on the HP1 protein. Generally the methods used are sound providing a more in depth understanding of HP1 functions in responds of DNA damage induction.
This paper has a potential to be accepted, however, some important points have to be clarified or fixed.
1. The major caveat of the paper is the use of late time points to look at DNA Damage signalling. Signalling via gamma-H2AX and 53BP1 is fast and takes place up to the first 30 minutes upon damage in a normal background. Looking late will lead to variable and inconclusive results as many breaks would have been already repaired. Additionally, 5Gy dosing is too high for such experiments as it will induce cell cycle checkpoints faster (in fact it can clearly be seen in the westerns in Fig. 6d when S10-H3 is down). This reviewer would recommend a 2Gy dose and 10 minutes time point for initial signalling experiments. As such, in the paper it can be clearly seen a disconnect between the immune fluorescence data and western data; it is obvious from Fig. 6d that the signalling is already surpassed (no clear increase in gamma-H2AX) most likely because repair is already ongoing and cells are arrested in G2.
2. It would be of interest to see an additional group SAHA+IR treatment; this would provide a lot of important information.
3. In the mass spectrometry result section writing is lengthy and most should be transferred to methods. Importantly, the authors DO NOT provide the mass spec results; as per international publishing policy this data needs to be presented as well as provided to a freely available database. In this reviewer opinion, the paper CANNOT be published without presenting this data as readers need to see the results. “Data not shown” I don’t believe is acceptable anymore.
4. Pertinent to Figure 3a the authors make the observation that “Due to the results showing an increased HP1β-S88ph in γ-irradiated HeLa cells, and these cells treated by SAHA (statistically nonsignificant results) (Fig. 3a), we addressed a question if HDAC inhibitor, SAHA induces DNA damage.” This reviewer needs to emphasize that one cannot state that a p=0.06 is not significant just because is not smaller that p=0.05. As the authors I hope can appreciate the 5% confidence interval is marginally close to 6%. What this data shows is that authors need to do more repeats to increase power as to determine if indeed there is a difference or not. As it stands I would say that treatment with SAHA is as significantly different than IR.
5. Generally, the writing is too broad with the introduction more like a review than focussing on the aspects clearly important for this paper. Methods are extensive but sometimes miss important bits (reference or show the construct of mCherry-pBABE puro vector). The full name of the HDAC inhibitor is mentioned in methods but not the main text; it should be added when first appears.
6. The Figures generally need additional details as well as quantification of immune fluorescence data. All figures that have treatments need to indicate on the figure as well as figure legend compound dose and time post treatment.
7. Discussion needs to be revised, as again it reads away from discussing the results and more like a review; it should be directly related to the work. For example, there was no note of Ku70 in introduction or results but is being discussed.
Author Response
We thank reviewers for their suggestions on how to improve our manuscript. Below, we reply to each of the reviewers' criticism. In the revised manuscript, text changes are denoted with red fonts. Also, we would like to confirm that our manuscript was written according to the instructions provided by Cells journal, for authors. Also, we would like to acknowledge that our work has not been published previously, and the manuscript is not under consideration for publication elsewhere. This text was approved by all authors.
Reviewer 1st
The manuscript by Legartova S et al. explores the relationship between sensing/accumulation of DNA Damage and the regulation of the chromatin environment, focussing on the HP1 protein. Generally the methods used are sound, providing a more in-depth understanding of HP1 functions in response to DNA damage induction.
This paper has a potential to be accepted, however, some important points have to be clarified or fixed.
The major caveat of the paper is the use of late time points to look at DNA Damage signalling. Signalling via gamma-H2AX and 53BP1 is fast and takes place up to the first 30 minutes upon damage in a normal background. Looking late will lead to variable and inconclusive results as many breaks would have been already repaired. Additionally, 5Gy dosing is too high for such experiments as it will induce cell cycle checkpoints faster (in fact it can clearly be seen in the westerns in Fig. 6d when S10-H3 is down). This reviewer would recommend a 2Gy dose and 10 minutes time point for initial signalling experiments. As such, in the paper it can be clearly seen a disconnect between the immune fluorescence data and western data; it is obvious from Fig. 6d that the signalling is already surpassed (no clear increase in gamma-H2AX) most likely because repair is already ongoing and cells are arrested in G2.Answer: In the revised version, we performed additional western blots to show the effects of a lower dose of γ-irradiation and the shorter period after irradiation. And also, a combination of γ-irradiation together with SAHA treatment, as the reviewer suggested. By western blots, we compared the non-irradiated control samples with 5 Gy of gamma-rays (cells were harvested 2 hours after irradiation), 2 Gy (cells were harvested 10 min after irradiation), 2 Gy (cells were harvested 30 min after irradiation), and 2 Gy (cells were harvested 10 min after irradiation) + SAHA, 2 Gy (cells were harvested 30 min after irradiation) + SAHA. The final concentration of HDACi (SAHA) was 15 μM, and this concentration we used for the cell treatment; cells were harvested 2 hours after the treatment. By western blot (see revised Fig. 8d), we analyzed the following markers: HP1α, HP1β, and HP1γ, γH2AX, H3S10ph. Western blot data we quantified using the ImageJ 1.32 software and data are shown in the revised manuscript in Fig. 8d, e.
Additionally, we analyzed the volume and a number of γH2AX foci in non-irradiated cells, cells exposed to 5 Gy of gamma rays (2 hours), 2 Gy (30 min), and 2 Gy (30 min) + SAHA (Fig. 9), as well as we, analyzed the volume and number of γH2AX- and 53BP-positive positive nuclear foci in non-irradiated cells, 5 Gy (2 hours) and SAHA treated cells (Fig. 5a-e). This analysis was performed by using ImageJ 1.32 software, which enables to find the individual DNA damage related foci automatically and software calculate the volume of foci studied by 3D-confocal microscopy.
It would be of interest to see an additional group SAHA+IR treatment; this would provide a lot of important information.Answer: See our 1st answer and a revised Fig. 8d, e, and 9.
In the mass spectrometry result section writing is lengthy and most should be transferred to methods. Importantly, the authors DO NOT provide the mass spec results; as per international publishing policy this data needs to be presented as well as provided to a freely available database. In this reviewer opinion, the paper CANNOT be published without presenting this data as readers need to see the results. “Data not shown” I don’t believe is acceptable anymore.Answer: We uploaded mass spec data into the PRIDE database. We added corresponding link and reference into the manuscript: “The mass spectrometry proteomics data have been deposited to the ProteomeXchange Consortium via the PRIDE (Vizcaíno et al., 2016) partner repository with the dataset identifier PXD014802.”Reviewer account details: Username: reviewer09641@ebi.ac.uk Password: gvv1yI0fData presented as “not shown” were added as Supplementary Table 1 in the revised version of the manuscript. Result section on mass spec was shortened in the revised version of the manuscript; we followed the Reviewer’s comment and transferred part of the text to the Methodology.
Pertinent to Figure 3a the authors make the observation that “Due to the results showing an increased HP1β-S88ph in γ-irradiated HeLa cells, and these cells treated by SAHA (statistically nonsignificant results) (Fig. 3a), we addressed a question if HDAC inhibitor, SAHA induces DNA damage.” This reviewer needs to emphasize that one cannot state that a p=0.06 is not significant just because is not smaller that p=0.05. As the authors I hope can appreciate the 5% confidence interval is marginally close to 6%. What this data shows is that authors need to do more repeats to increase power as to determine if indeed there is a difference or not. As it stands I would say that treatment with SAHA is as significantly different than IR.Answer:
We completely agree with the Reviewer’s comment that both p-value and statistical power depend on the variability and the number of replicates, and that p=0.06 is closed to the level of significance. So, we can only confirm the claim of the Reviewer that the treatment with SAHA is almost as significantly different as IR, and what seems to be valuable information that SAHA treatment (in the concentration that we used for hyperacetylation and chromatin relaxation) does not induce DNA damage significantly. From the view of HP1β-S88ph, we can additionally say that this is the post-translation modification of HP1b that seems to be sensitive to both irradiation and hyperacetylation, the processes with a potential to significantly change chromatin structure.
To follow Reviewer’s comment, we rewrote the text concerning SAHA treatment as follows:
“Compared to the non-irradiated control samples, the level of G81-E96 peptide carrying phosphorylation at serine 88 (m/z 596.5863+++; HP1β-S88ph) in γ-irradiated samples increased significantly (p = 0.02; Fig. 4a and 6a, b). Approaching the borderline of significance and more than 1.5-fold increase in the level of this peptide was also found in the cells treated by clinically used HDAC inhibitor, SAHA (p = 0.06; Fig. 4a and 6a).”
Generally, the writing is too broad with the introduction more like a review than focussing on the aspects clearly important for this paper. Methods are extensive but sometimes miss important bits (reference or show the construct of mCherry-pBABE puro vector). The full name of the HDAC inhibitor is mentioned in methods but not the main text; it should be added when first appears.Answer: We revised the Introduction section, and we showed reference for mCherry-pBABE puro vector, and the full name of HDAC inhibitor SAHA, we explained in the main text (in this case we thought that is not necessary to introduce SAHA, as well-known, clinically used HDACs inhibitor.
The Figures generally need additional details as well as quantification of immune fluorescence data. All figures that have treatments need to indicate on the figure as well as figure legend compound dose and time post treatment.Answer: We revised figure legends and wrote doses and periods of analyses directly to the figure legends. We performed quantification of the volume and the number of gH2AX-53BP1-positive nuclear foci in non-irradiated cells, cells exposed to 5 Gy of gamma rays (2 hours) and SAHA treated cells, see revised Fig. 5. The analysis was performed by using ImageJ 1.32 software, which enables to automatically find the individual DNA damage related foci and calculates their numbers and volume.
In Fig. 6b (revised 7b, and it is also Fig. 1) we quantified foci of HP1 isoforms in the vicinity of nucleoli (Fig. 1d), but additionally, we performed more precise analysis by ChIP-PCR. We analyzed an abundance of HP1α, HP1β, HP1γ, and KAP1 in ribosomal genes (promoter of rDNA, and DNA encoding 28S rRNA). Importantly, we observed a radiation-induced reduced level of HP1 alpha in both rDNA encoding 28S rRNA and rDNA promoter region (Fig. 2a, b). ChIP-PCR additionally showed, in comparison to HP1α and HP1β the highest level of HP1γ in ribosomal genes, and especially in rDNA promoter region. Importantly, γ-irradiation did not affect the level of HP1γ in ribosomal genes (Fig. 2a, b).
Discussion needs to be revised, as again it reads away from discussing the results and more like a review; it should be directly related to the work. For example, there was no note of Ku70 in introduction or results but is being discussed.Answer: Discussion was revised and, and a little information about the function of Ku70 factor we deleted from the Discussion section due to the fact that this DNA repair-related factor was not in the focus of our studies.

Reviewer 2 Report
Comments:
1. P7, in the Result 3.1 section: the authors described that gamma-radiation reduced the number and the size of HP1alpha-positive foci in the vicinity of nucleoli for Fig.1a. It will be more convincing if the authors could quantify from for example 50-100 cells, compare cells with and without gamma-radiation treatment and do the statistical analysis to confirm.
2. P14, in the Result 3.3 section: the authors did not give enough introduction of the design of the IPs in Fig.6. It is also not clear in the method section that how the co-IP detection of, for example HP1alpha-HP1alpha interaction, works in the setting of this IP experiment. The WB of the HP1alpha in Fig.1a, for example, seems not the pulldown of the endogenous HP1alpha (the authors claimed it was the co-IPed HP1alpha, which do not make sense) by the IP of HP1alpha antibody. Please explain in details in order to not confuse the readers.
3. P14, in the discussion paragraph 1: for a solid study, what you have in your mass spectrum dataset need to confirm by other traditional methods, for example, WB or IF. I understand that it might be difficult to generate a HP1beta-S88P antibody for the traditional experiment to confirm your conclusion. It is better to tone down a little bit in your conclusion about HP1beta-S88P increasing after gamma-radiation because the conclusion is only based on MS data. On the contrary, what you missed in your mass spectrum dataset could not support that it does not exist in this cell line. Therefore, it is not fair to say that “post-translational modifications of HP1 isoforms are cell-type specific” only based on your MS data for HeLa and LeRoy’s MS data for another cell line.
4. Minor: the authors wrote several times “by [Ref.]”. It is better in “by Surname et. al. [Ref.]”.
Author Response
We thank reviewers for their suggestions on how to improve our manuscript. Below, we reply to each of the reviewers' criticism. In the revised manuscript, text changes are denoted with red fonts. Also, we would like to confirm that our manuscript was written according to the instructions provided by Cells journal, for authors. Also, we would like to acknowledge that our work has not been published previously, and the manuscript is not under consideration for publication elsewhere. This text was approved by all authors.
Reviewer 2nd
P7, in the Result 3.1 section: the authors described that gamma-radiation reduced the number and the size of HP1alpha-positive foci in the vicinity of nucleoli for Fig.1a. It will be more convincing if the authors could quantify from for example 50-100 cells, compare cells with and without gamma-radiation treatment and do the statistical analysis to confirm.Answer: In Fig. 6b (revised 7b, and it is also Fig. 1) quantified foci of HP1 vicinity of nucleoli, but we also performed more precise analysis by ChIP-PCR. We analyzed an abundance of HP1α, HP1β, HP1γ, and KAP1 in ribosomal genes (promoter of rDNA, and DNA encoding 28S rRNA). Importantly, we observed a radiation-induced reduced level of HP1 alpha in both rDNA encoding 28S rRNA and rDNA promoter region (Fig. 2a, b). ChIP-PCR additionally showed, in comparison to HP1α and HP1β the highest level of HP1γ in ribosomal genes, and especially in the rDNA promoter region. Importantly, γ-irradiation did not affect the level of HP1γ in ribosomal genes (Fig. 2a, b).
According to our experiences, ChIP-PCR has better informative value than the counting of foci in the vicinity of nucleoli. Moreover, we wanted to know in which extent HP1 isoforms occupy ribosomal genes and if gamma-radiation can change the abundance of HP1s in ribosomal genes.
P14, in the Result 3.3 section: the authors did not give enough introduction of the design of the IPs in Fig.6. It is also not clear in the method section that how the co-IP detection of, for example HP1alpha-HP1alpha interaction, works in the setting of this IP experiment. The WB of the HP1alpha in Fig.1a, for example, seems not the pulldown of the endogenous HP1alpha (the authors claimed it was the co-IPed HP1alpha, which do not make sense) by the IP of HP1alpha antibody. Please explain in details in order to not confuse the readers.Answer: For Immunoprecipitation experiments, we are using Catch and Release®v2.0 Reversible Immunoprecipitation System. It is based on (instead of use Protein A or Protein G coupled with agarose beads, to capture an antigen: antibody complex in solution) the Columns that contain a proprietary resin in a microfuge-compatible Spin Column secured by a screw cap top and a breakaway closure on the bottom. The binding between the resin and antigen: antibody complex is mediated via Antibody Capture Affinity Ligand.
The low pull-down of the endogenous HP1alpha in IPs data could be caused by unknown design and the epitope of commercially provided antibodies. For verifying results from IPs, we are using the FLIM-FRET technique. Moreover, fluorescent protein-technology, used for FLIM-FRET, enables a clear insight into the functional features of the cell nucleus, due to the analysis of the levels of exogenous protein. To be sure with protein-protein interaction, we are using the following rule: for interaction partners, we consider only proteins confirmed by both methods, IPs and FLIM-FRET. In the case of both IPs and FLIM-FRET negativity, we consider that these proteins work independently.
P14, in the discussion paragraph 1: for a solid study, what you have in your mass spectrum dataset need to confirm by other traditional methods, for example, WB or IF. I understand that it might be difficult to generate a HP1beta-S88P antibody for the traditional experiment to confirm your conclusion. It is better to tone down a little bit in your conclusion about HP1beta-S88P increasing after gamma-radiation because the conclusion is only based on MS data. On the contrary, what you missed in your mass spectrum dataset could not support that it does not exist in this cell line. Therefore, it is not fair to say that “post-translational modifications of HP1 isoforms are cell-type specific” only based on your MS data for HeLa and LeRoy’s MS data for another cell line.
Answer: We agree with the reviewer that it is not optimal to say that “post-translational modifications of HP1 isoforms are cell-type specific” only based on your MS data for HeLa and LeRoy’s MS data for another cell line. We can only say that our results by mass spec showed that HP1 modifications differ from those published by LeRoy et al. (2009). This claim we corrected in the revised version.
Note: indeed it would be really tricky to get antibody against S88ph-HP1b, so we can only use our mass spec data, similarly as it was published by LeRoy et al. (2009) that also did not confirm all described modifications of HP1 isoforms by appropriate antibodies, used for western blot analysis.
Minor: the authors wrote several times “by [Ref.]”. It is better in “by Surname et. al. [Ref.]”.
Answer: This form of citation was corrected. In the original manuscript, we used a form generated by EndNote software that is using a style recommended by a given journal; thus, we used the style of journal, called Cells. Also, we think that the style used by EndNote is not optimal, so, in the revised version, we apply the style recommended by the reviewer. It is much better.
